# Dyes from the Ashes: Discovering and Characterizing Natural Dyes from Mineralized Textiles

**DOI:** 10.3390/molecules25061417

**Published:** 2020-03-20

**Authors:** Alessandro Ciccola, Ilaria Serafini, Francesca Ripanti, Flaminia Vincenti, Francesca Coletti, Armandodoriano Bianco, Claudia Fasolato, Camilla Montesano, Marco Galli, Roberta Curini, Paolo Postorino

**Affiliations:** 1Chemistry Department, “Sapienza” “Sapienza” University of Rome, Piazzale Aldo Moro 5, 00185 Rome, Italy; alessandro.ciccola@uniroma1.it (A.C.); ilaria.serafini@uniroma1.it (I.S.); flaminia.vincenti@uniroma1.it (F.V.); armandodoriano.bianco@uniroma1.it (A.B.); camilla.montesano@uniroma1.it (C.M.); roberta.curini@uniroma1.it (R.C.); 2Department Physics, “Sapienza” University of Rome, Piazzale Aldo Moro 5, 00174 Rome, Italy; ripantifrancesca@gmail.com; 3Department Ancient Sciences, “Sapienza” University of Rome, Piazzale Aldo Moro 5, 00185 Rome, Italy; francescacoletti1986@gmail.com (F.C.); marco.galli@uniroma1.it (M.G.); 4Department Classical Archaeology, Ruprecht-Karls-University of Heidelberg, Grabengasse 1, 69117 Heidelberg, Germany; 5Department Physics and Geology, University of Perugia, Via Alessandro Pascoli, 06123 Perugia PG, Italy; claudia.fasolato@unipg.it

**Keywords:** Pompeii, Tyrian purple, shellfish purple, SERS, HPLC-HRMS, orbitrap, cultural heritage, mineralized textiles

## Abstract

Vesuvius eruption that destroyed Pompeii in AD 79 represents one of the most important events in history. The cataclysm left behind an abundance of archeological evidence representing a fundamental source of the knowledge we have about ancient Roman material culture and technology. A great number of textiles have been preserved, rarely maintaining traces of their original color, since they are mainly in the mineralized and carbonized state. However, one outstanding textile sample displays a brilliant purple color and traces of gold strips. Since the purple was one of the most exclusive dyes in antiquity, its presence in an important commercial site like Pompeii induces us to deepen the knowledge of such artifacts and provide further information on their history. For this reason, the characterization of the purple color was the main scope of this research, and to deepen the knowledge of such artifacts, the SERS (Surface Enhanced Raman Scattering) in solution approach was applied. Then, these data were enriched by HPLC-HRMS analyses, which confirmed SERS-based hypotheses and also allowed to hypothesize the species of the origin mollusk. In this context, a step-by-step integrated approach resulted fundamental to maximize the information content and to provide new data on textile manufacturing and trade in antiquity.

## 1. Introduction

In recent years, studies in archeological sciences and analytical chemistry have been moving towards the same direction. Indeed, due to the complex nature of archeological samples, their chemical and physical analysis inevitably requires a multi-analytical and multi-technical approach. Furthermore, considering the uniqueness of these samples and the difficulty in producing laboratory mock-ups to test novel analytical protocols, the scientific research requires advances in the standard analytical techniques, pushing their limits in terms of sensitivity and efficiency, especially in the case of materials of unknown composition.

Ancient textiles have gained great relevance in the field of archeology as a material of considerable potential to reconstruct crucial aspects of social, technological, and economic factors of past societies [1,2,3,4,5,6,7,8,9,10]. The growing research into archeological textile materials has led to taking of unexplored or scarcely investigated collections into account. Pompeii may represent the most important case study, which perfectly fits with the definition of this new challenge. In fact, the wide textile collection of Pompeii offers one of the major opportunities to apply and test more advanced archeological and scientific techniques. In this context, the investigation of archeological textile samples is a great challenge due to the highly perishable nature of organic materials.

Among these, major importance is attributed to the study of ancient dyes and, in particular, to shellfish purple. Some mollusk species were traditionally used in Mediterranean regions to produce such a precious and expensive colorant for dyeing textiles in antiquity. Moreover, this dye was used in connection with gold thread or strips in order to raise the value and the price of fabrics, respectively [11]. Since the Bronze Age and throughout the antiquity, in fact, purple textiles have been used as a mark of social and economic status, as well as of opulence and prestige. During Roman times, purple dye, likely in the violet hue, was the color of the senatorial and equestrian rank, and later of the Emperor’s family only [12]. The different color shades range from red and purple to violet-blue [13]. The process of dye extraction from *Murex* mollusks and the chemical reactions involved in the formation of different purple hues are considerably more complex than what is required to produce organic dyes from vegetal sources [14], and they represent an example of elaborated manufacturing processes in antiquity.

Currently, researchers demonstrate how some advances in the identification of shellfish purple have been reached by studying chemical composition of the purple dye obtained from animal matrices and textiles [15,16]. In particular, several in-depth studies have been dedicated to investigate the dyestuff components of the shellfish-based dyes and the dyeing process techniques in order to advance our knowledge about purple dye production in antiquity: primary sources (mollusk species), extraction methods, dye processing, final shade color range [13,17,18,19]. According to these studies, the purple dye was extracted from several sea snails belonging to the Muricidae family that secrete the coloring matter from their hypobronchial gland after their death [13,20]. “True Purple” dyes were mainly produced from three species of Mediterranean Sea mollusks: *Hexaplex trunculus* (L.) (*Murex trunculus*), *Bolinus brandaris* (L.) (*Murex brandaris*), and *Stramonita haemastoma* (L.) (*Thais haemastoma*) [21].

The identification of the dye source of a textile is a complex issue, which implicates great attention to several factors. Reported cases of chemical identification of shellfish purple dyes in archeological samples are not common. The analyses performed on the mollusks and reported in literature show variations in composition of the extracts from different shellfish species [15,16,18]. In the past, identification of the species was often conducted with reference only to the content of monobromoindigotin and dibromoindigotin. Actually, according to the analyses reported in literature, the largest content of non-brominated compounds could be considered indicative of *Hexaplex trunculus*, while the other two species present a larger amount of mono- and dibrominated dyes. Certainly, the concentration of the different compounds in the extract from the animal source could be not representative of the composition of the dye on the textiles: it was shown, for instance, that monobrominated indigotin presents higher affinity for wool fibers in wat dyeing processes in comparison to the other compounds [22]. Furthermore, the applied dyeing procedure has to be taken into consideration: it was shown, for instance, that monobrominated indigotin in its leuco form is highly sensitive to visible light at some pH values, and it is prone to debromination, so the exposure of the dye bath to light could cause variations in the colorants effectively fixed on the fiber [22]. Finally, ageing effects have to be considered even if, according to the literature, they should not constitute the main factor responsible for variations of the dye mixture on the textile: high temperatures could be responsible for color variations even if they should not correspond to chemical degradation of the dyes fixed on the fiber [23], while no great variation was observed in relative concentrations of the dyes before and after UV exposure [24]. All these aspects are stressed out in the literature, where identification of specific mollusk species is often only hypothesized.

In this study, a multi-technical approach is applied to one mineralized textile fragment belonging to the MANN (Museo Archeologico Nazionale di Napoli) collection (Figure 1). The analyses were performed on one fabric decorated with gold with evidence of a purple dye (the factors which suggest it constituted a luxurious item) preserved in the mineralized form. The main focus of this paper is to discuss the in-depth scientific analyses carried out on this colorant in order to identify the major dye components or molecular markers and to propose a hypothesis about the *Murex* species that was used taking into account the current scientific discussion about natural sources of purple dyes, the informative content of the adopted scientific approach, and the complexity of the issue [15,22,23,24,25].

The investigation’s focus is based on the international scene of scientific research, which has recently demonstrated high interest towards the interdisciplinary research related to the study and identification of the purple dye sources and manufacture in antiquity [16,25,26,27,28,29]. The reported results must be considered useful to provide new data to the actual knowledge on shellfish purple and to push forward the research on the identification of specific natural sources from analyses of textiles.

In the end, this information obtained from unique samples such as those presented in the paper can therefore fit into the wider context of the site’s historical-archeological reconstruction and also stimulate construction of a bridge between archeology and science, whose bases are now increasingly solid.

## 2. Results

### 2.1. Microscopic Techniques

In order to successfully investigate this found textile, it was fundamental to evaluate the preservation state of the fibers in order to obtain preliminary information from its optical appearance. Consequently, the more appropriate investigation methodology was selected. The technical analyses of the textile surface were performed using a traditional optical tool. At the same time, since the textile is mineralized, the morphological analyses of the fibers were only possible by the use of a scanning electron microscope (SEM). The analyses showed the complete alteration of the fiber morphology (Figure 2a) and the presence of some gypsum crystals (Figure 2b).

However, some considerations about the type of the fibers used to manufacture the items can be given. The textile surface displays fine threads with a diameter of 0.15 mm, z-twisted at an angle of 35–50°, observable using a stereo microscope (Figure 2c). The visible sections of the gold strips show a small thickness of 1–2 μm. In this case, the state of preservation of the sample does not allow for the identification of the type of gold inserts as well as of the type of interaction between the gold and the purple threads. However, the archeological evidence, along with the esthetical characteristics of the objects and the materials appearance, suggests that this fabric may be regarded as an example of purple and gold tapestry [30,31]. The praxis in antiquity to dye animal fibers (wool and silk), usually in the form of fleece [22,32], was based on their natural predisposition to better binding with natural dyes in contrast to cellulose fibers. The direction of z-twisted threads is traditionally linked to wool yarn in the ancient European/Mediterranean spinning practices as opposed to silk (no visible twist (“I-twist”)) and plant-based (s-twisted) fibers.

Based on these data, it can be assumed that the mineralized textiles found in Pompeii was made of woolen fibers and decorated with gold, while further analyses were considered for the identification of the purple dye.

### 2.2. Raman and SERS Analysis

In Figure 3, we present the results of the Raman and SERS spectroscopic study of the present sample, along with the reference data from the literature for their interpretation. The Raman spectrum of the sample (Figure 3, top panel, spectrum (a)) presents the typical features of indigoid dyes reported in literature [33,34,35,36]. In particular, the peaks attributable to C=C and C=O stretching are visible between 1550 and 1710 cm^−1^, the most intense being centered at 1582 cm^−1^. In the range between 1300 and 1450 cm^−1^, the signals attributable to N-H and C-H bendings are observable, while at low wave numbers (900-1250 cm^−1^), all the signals of bending and deformation of C-C and C-H groups are revealed. In the 250-800 cm^−1^ region, several weak bands can be observed and attributed to bendings and deformations of two-, three-, and four-atom groups. The strong signal at 309 cm^−1^ can be considered indicative of the bromine substituent on the Indigotin scaffold.

The comparison with the Raman spectra reported in literature (Figure 3, central panel) evidences the complex composition of the considered class of dyes. In particular, if there is a general matching with the Raman spectra of indigotin (spectrum (f)) and 6,6’-dibromoindigotin (spectrum (e)) acquired with the same laser wavelength [35], some bands feature slightly different intensities. In particular, the peak at 1254 cm^−1^ attributed to C-H and C=O bendings is less intense in comparison to its analogue in the spectrum of pure 6,6’-dibromoindigotin. On the contrary, the peak at 309 cm^−1^ is more intense in the case of the sample spectrum in comparison to the dibromoindigotin one. Furthermore, it is worth noticing that in the regions between 1470–1760 and 1260–1425 cm^−1^, the resulting signals are broader and overlapping. This set of observations could suggest that the different markers of Tyrian purple (non- and monobromoderivatives) are likely present in the mixture in considerable amounts as also evidenced by Table 1 which compares the characteristic peak in the Raman spectrum of the sample to those of some characteristic markers reported in literature with reference to their tentative attributions.

In order to obtain more information about the total composition of the dye mixture, the surface-enhanced Raman scattering (SERS) analysis was applied to the characterization of the dimethylformamide (DMF) extract from the samples. The SERS approach is based on the contact between an analyte and noble metal nanostructures, which provides a great enhancement of Raman signals from electromagnetic and charge-transfer mechanisms [37]. The solution was analyzed in order to emphasize the signals of all the compounds extracted in a homogenous environment minimizing the effects related to different adsorption to the nanoparticles in the solid state, which could affect the SERS spectral shape and affect or hamper the molecular identification [38].

The application of the SERS in solution analysis allowed obtaining high-quality spectra with a very high signal/noise ratio (Figure 3b); furthermore, the spectra from repeated acquisitions featured a very reproducible spectral shape, which was not affected by the contribution of the colloid or solvent background, as evident from the comparison with the reference spectrum (Figure 3c).

As the Raman signal, the SERS spectrum shows the main peaks of indigoids and, specifically, resembles the conventional Raman signature of 6,6’-dibromoindigotin (e). Moreover, it is important to notice that a great enhancement in the relative intensity can be observed for the high-frequency peaks (1457, 1415, 1360, 1247, 967, and 612 cm^−1^), whereas the signal at 309 cm^−1^ is hardly visible. The absence of this low-frequency signal suggests no main interaction of the bromine substituent with the nanoparticle; moreover, the peaks between 1240 and 1460 cm^−1^, as well as the two peaks at 612 and 967 cm^−1^, are likely attributable to N-H and C-H modes, suggesting a higher interaction of these groups with the nanoparticle surface. As also discussed in the literature [39], this observation could be attributed to a large amount of non- and monobrominated compounds present in the Tyrian purple mixture. Eventually, we would stress the high quality of the SERS spectrum acquired in our sample compared to other spectra of indigotin (Figure 3, spectrum (h)) and Tyrian purple present in the literature [39]. The present data could represent a well-identifiable reference SERS spectrum for this dye mixture.

### 2.3. HPLC-HRMS Analyses

The data obtained by HPLC-HRMS of the extracts from mineralized dyes allowed to identify the main colorants, typical of precious Tyrian purple, such as monobromoindigoid isomers and non-brominated ones, indigotin and indirubin (Figure 4). Taking into account the presence of several isomers in the dye, it is important to highlight that only three chromatographic peaks were observed corresponding to the accurate masses of non-brominated, monobrominated, and dibrominated indigoids. Relative quantification was obtained by calculating the ratio between the area of the peak of interest and the sum of the area of all the peaks. Following this approach, the amounts of compounds of different bromination grade were compared. In particular, the dyeing mixture seems to be composed of 80% non-brominated species, likely indigotin, while 19% is represented by monobromo compounds and 1% by dibromo compounds, which should correspond to monobromoindigotin and dibromoindigotin.

## 3. Discussion

The possibility of investigating archeological samples imposes the use of the analytical and diagnostic techniques as versatile as possible. In addition, the uniqueness of these findings force us to use highly sensitive techniques that ensure excellent reproducibility and robustness of the results. For this reason, in terms of sensitivity, a step-by-step approach was chosen that took into account some non-invasive and non-destructive analyses, such as microscopic techniques and Raman spectroscopy, followed by micro-invasive SERS spectroscopy and, finally, by chromatography and mass spectrometry, which provided the final analytical confirmation.

Despite the small size and the degree of degradation (no traces of the original organic thread is left, while some alteration of the morphology occurred, as evidenced previously), this fragment preserved the original purple dye and traces of gold strips (Figure 2b–d). Moreover, the mineralization process stopped the natural biodegradation of the organic material. This process favored a progressive substitution of the sample’s organic matrix with an inorganic one, forming a replica of the textile material [29,41]. The fragment likely maintained its color due to the chemical stability of the shellfish purple [29].

According to the direct Raman analysis, it was clear that the dye present on the mineralized textile was an indigoid purple dye. However, some deviations from the spectrum of pure 6,6’-dibromoindigo suggested that other compounds could be present in a significant amount. Unfortunately, the high fluorescence in several areas did not allow acquiring a large number of spectra, whose analysis could result useful to confirm the semi-quantitative results for the dye distributed on the whole sample. In order to prove the presence of other considerable molecular species, other techniques were adopted. In particular, the SERS analysis of the extract was considered useful to obtain the information representative of the overall composition of the dye: in fact, high reproducibility of SERS spectra in the solution and no interference from colloid signals granted the acquisition of informative spectra which allowed hypotheses about the presence of the molecular species different from a dibrominated compound. In particular, SERS signals suggested that non- and monobrominated compounds were present: as evidenced by Bruni [39], a smaller content of bromine atoms could be responsible for the remarkable interaction of these species with the nanoparticles, facilitating the contact of N-H and C-H groups with the Ag surface. However, it is appropriate to consider the limitations of this phenomenon. The preferential surface interaction of non- and monobrominated compounds on nanoparticles with respect to dibromoindigo is only indicative of their presence in the mixture, but no conclusive semi-quantitative consideration can be derived: dibromoindigoids could be the most abundant species in the dye mixture, but the major affinity of the other compounds to Ag nanoparticles could minimize their SERS detection. From this point of view, HPLC-HRMS resulted fundamental in order to finally confirm the presence of less brominated compounds and to estimate their relative concentration. Mono- and non-brominated species were identified successfully on the basis of their accurate masses, and a semi-quantitative analysis was performed according to the method discussed in Section 2.3. Even if differences in the matrix effect and/or ionization efficiency may lead to differences in ion intensity, non-brominated species appear to be the most abundant compounds, followed by monobromo compounds, while dibromo compounds have a lower abundance. These results confirm Raman- and SERS- based hypotheses. In particular, the non-brominated compounds present in the dye should be indigotin and indirubin, which are isomers. However, a single chromatographic peak was observed for the relative value of accurate mass. Considering the relative concentrations typically reported in the literature on extracts from mollusks and from textiles [15,16], it can be stated that the amount of indirubin is often very low in comparison to indigotin. Moreover, the Raman spectrum does not show specific peaks for indirubin. Based on these considerations, it can be hypothesized that the dominant non-brominated species is indigotin, while the indirubin concentration is likely below the limit of detection. The alternative hypothesis, i.e., that the separation was not complete, seems unlikely taking into account the structural differences between the two compounds. Finally, according to semi-quantitative analyses and the comparison with literature data, further considerations were derived about the origin of the natural matrix used for dye extraction. In fact, the high content of indigotin is generally considered indicative of the use of a *Hexaplex trunculus* (L.) mollusk as a dye source [16,17]: the decreasing content of non-, mono-, and dibrominated compounds is characteristic of this species, while *Bolinus brandaris* (L.) and *Stramonita haemastoma* present dibromoindigotin and dibromoindirubin as the most abundant molecular markers, with only traces or even no content of non- and monobrominated compounds [18]. It is correct to highlight that the relative amount of indigotin and indirubin in the analyzed Pompeii sample is also higher (80%) in comparison to the data reported in literature for mollusk extracts (maximum 64%) and other archeological samples (maximum 48%) [16]. This higher concentration could suggest that a specific class of *Hexaplex trunculus* nails was used: in fact, according to literature [21], in nature, there are indigo-poor and indigo-rich exemplars of this mollusk species, and the latter ones are likely the original matrix used for the production of the dyes in the sample. However, it is important to highlight that in the reported work, the dye was extracted from a mineralized textile with a complex history, and not directly from a mollusk dye bath, so the abundance in a non-brominated compound could be related to other factors: for instance, it could be a consequence of the dyeing technology or exposure of the dyes to sun [22,23]. Finally, the influence of high temperatures connected to the Vesuvius eruption on dye degradation cannot be excluded even if, considering some literature studies based on mock-ups, no definitive evidence of chemical degradation induced by temperature was observed [23]. Taking into account the unique character of the Pompeii case, however, this perspective would need further studies, which will constitute the next step of the research.

## 4. Materials and Methods

### 4.1. Materials and Instruments

Solvents and reagents were purchased from Fluka, Sigma–Aldrich, and Carlo Erba and used without further purification. For analyses by means of HPLC-HRMS, solvents were purchased from Fisher Scientific.

### 4.2. Mineralized Dyed Textile

The sample analyzed is part of a group of eight mineralized textiles, all found in Pompeii [25,37]. Their exact location is unknown, since it was not recorded during their 19th century excavation. The textile under investigation has the max. dimension of 3 × 1.3 x 0.6 cm (Figure 2a). The textile was produced in an extremely fine and balanced tabby weave with a supplementary gold ornamental element [42,43,44].

### 4.3. Extraction Procedure for Purple Dyes

Taking into account the morphology of the samples, which is mineralized, the purple dye layer appears as a compact brittle and powdery layer. For this reason, it was possible to remove up to 2 mg of the mineralized purple dye. Then, according to literature, two different extraction protocols were used, one suitable for SERS analyses, and one for HPLC-HRMS analyses.

First, the extraction protocol in DMF was applied. 1 mg of the purple powder was suspended in 1 mL of *N*,*N*-dimethylformamide and heated at 70 °C for 5 min [16,39]. Then, the extract was put in contact with an Ag colloid.

For the HPLC-HRMS analysis, the same protocol was applied, but 1 mL of dimethyl sulfoxide (DMSO) was used as a solvent because of its higher extracting power as reported in literature [16,21]. The extract was filtered through a PTFE filter (0.20 µm), and an aliquot of 10 µL was suspended in 200 µL of methanol.

### 4.4. Preparation of SERS Colloid

The Ag-reduced colloid was prepared according to the protocol developed by Leopold and Lendl [45]. Briefly, a solution of 1 × 10^−3^ M AgNO_3_ in the MilliQ water was prepared. Separately, the same volumes of a solution of 6 × 10^−2^ M NH_2_OH*HCl and a solution of 1 × 10^−1^ M NaOH were mixed together. 10 mL of the NH_2_OH*HCl solution were added to 100 mL of the AgNO_3_ solution under stirring, with direct formation of a colloid. The colloid was left under stirring for twenty minutes, and it was used in the next few days. Aggregation of the colloid was induced by adding 10 µL of the MgSO_4_ solution.

### 4.5. Raman and SERS Analyses

Raman and SERS analyses were performed using a Horiba Jobin–Yvon HR-Evolution micro-Raman setup equipped with a 632 nm laser. A motorized mapping stage was used for inspecting the sample and collecting Raman signals from specific locations on the sample.

The standard Raman analysis was performed directly on the purple area of the archeological sample. The set conditions for the spectral acquisition involved 50X magnification, laser power between 0.25 and 0.50 mW, maximum accumulation time of 10 s per scan, 60 scans maximum.

For the SERS in solution analysis, 100 µL of the Ag colloid and 50 µL of the DMF extract from the sample were added to an Eppendorf tube and stirred. Finally, 10 µL of the MgSO_4_ solution were added and stirred to induce aggregation of the nanoparticles. The final solution was poured in an analytical well on a plate, and spectra were acquired. The set conditions for the spectral acquisition involved 20X magnification, laser intensity of 1 mW, maximum accumulation time of 1 s per scan, 100 scans maximum. In order to evaluate eventual spectral interferences deriving from the presence of DMF, spectra were also collected from a blank solution prepared by mixing 100 µL of the Ag colloid, 50 µL of pure DMF, and 10 µL of the MgSO_4_ solution. Three spectra for every type of the analyzed solution (sample and blank) were acquired and averaged; the average spectrum of the sample was compared with the average one of the blank solutions in order to discriminate between the signals attributable to DMF and interfering signals.

### 4.6. HPLC-HRMS Analyses

HPLC-HRMS analyses were carried out using a Thermo Scientific Ultimate 3000 RSLC system coupled with a Thermo Scientific Q Exactive mass spectrometer (Thermo Fisher Scientific, Bremen, Germany). The Ultimate 3000 RSLC system consisted of a degasser, a tertiary loading pump, a binary eluting pump, a column oven, and an RS autosampler.

The HPLC analyses coupled with high-resolution mass spectrometry (HPLC-HRMS) were carried out using a Thermo Scientific Ultimate 3000 RSLC system coupled with a Thermo Scientific Q Exactive mass spectrometer (Thermo Fisher Scientific, Bremen, Germany). For the chromatographic separation, a Zorbax SB-Phenyl column was used (4.6 × 150 mm, 3.5 μm, 80 Å (Agilent Technologies)) together with a Zorbax SB-Phenyl precolumn (4.6 × 12.5 mm, 5.0 μm, Agilent Technologies). All the samples were solubilized in methanol prior to injection. The injection volume was 20 μL (Rheodyne model 7225i), and the flow rate was set at 0.5 mL min^−1^. The eluents were (A) 1.5% (v/v) formic acid in water, (B) methanol, following the gradient program (min. 0, A 60%; min. 14, A 40%; min. 20, A 30%; min. 27, A 0%; min. 30, A 0%). The stop time was 30.0 min., and the equilibrium time was 8.0 min. Detection was performed using a Q Exactive mass spectrometer equipped with a heated electrospray ionization source (HESI) with a PRM acquisition mode.

## 5. Conclusions

The present work combines highly sensitive analytical techniques for the characterization of ancient dyes in a mineralized textile from Pompeii and provides new information about textiles in the Western Mediterranean during the Roman period. The combination of Raman, SERS in solution, and HPLC-HRMS analyses resulted in the identification of purple dyes and the convergence of information obtained by the different techniques allowed the formulation of hypotheses about the specific mollusk species which constituted the natural matrix for colorant extraction. However, further research is necessary to improve the HPLC separation of different isomers in samples in order to achieve definitive confirmation of dye composition and to develop new HPLC-HRMS methods for this type of organic colorants. Finally, taking into account the achieved results and the thermochromic effects on purple dyes, this study opens new perspectives for the characterization of the thermal effects on organic materials during the Pompeii eruption.

From the archeological point of view, the mineralized textile analyzed offers clear evidence for the presence of luxury textiles made of gold and dyed with shellfish purple in Pompeii. Together with the high number of gold textiles found in the Vesuvian cities [30], this sample confirms the circulation of high-quality objects in Pompeii and the role of the ancient city as a textile hub, at least in terms of consumption. Purple-gold textiles were well-known in ancient times in the Eastern Mediterranean and in the western provinces of the Roman Empire during Late Antiquity [11]. Although Pompeii is a unicum in terms of preservation, such textile is remarkable considering that other contemporary and comparable archeological findings are lacking in the central area of the Roman Empire for the I century AD [11,46,47].

## Figures and Tables

**Figure 1 molecules-25-01417-f001:**
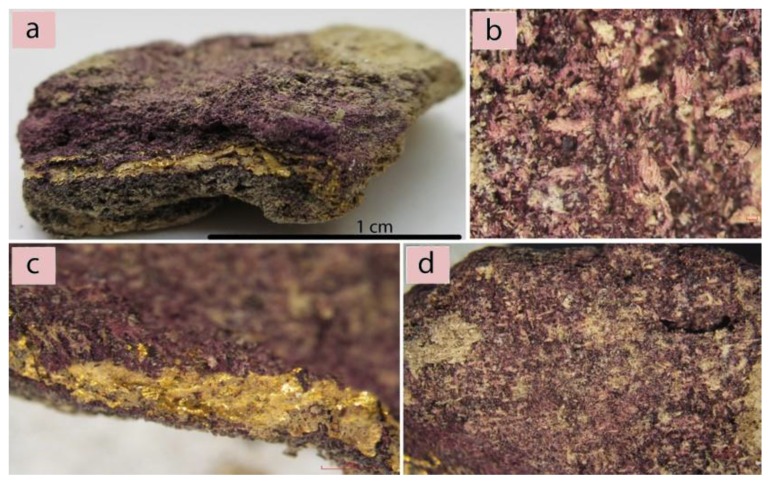
(**a**) The analyzed sample; (**b**) Mineralized structure of a textile with purple dye remains; (**c**) Purple dye and gold strip evidence; (**d**) A purple area on the surface of the sample.

**Figure 2 molecules-25-01417-f002:**
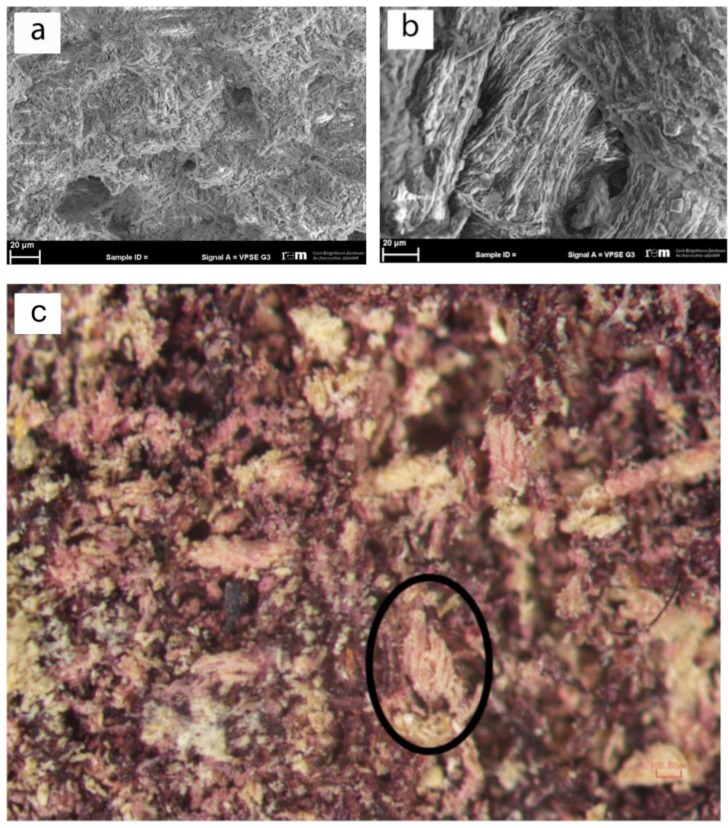
(**a**) SEM evidence of alteration of morphology of the sample’s fibers; (**b**) Presence of crystals with characteristic gypsum structures on altered fibers; (**c**) Stereo microscope image evidencing the z-twist feature (black circle).

**Figure 3 molecules-25-01417-f003:**
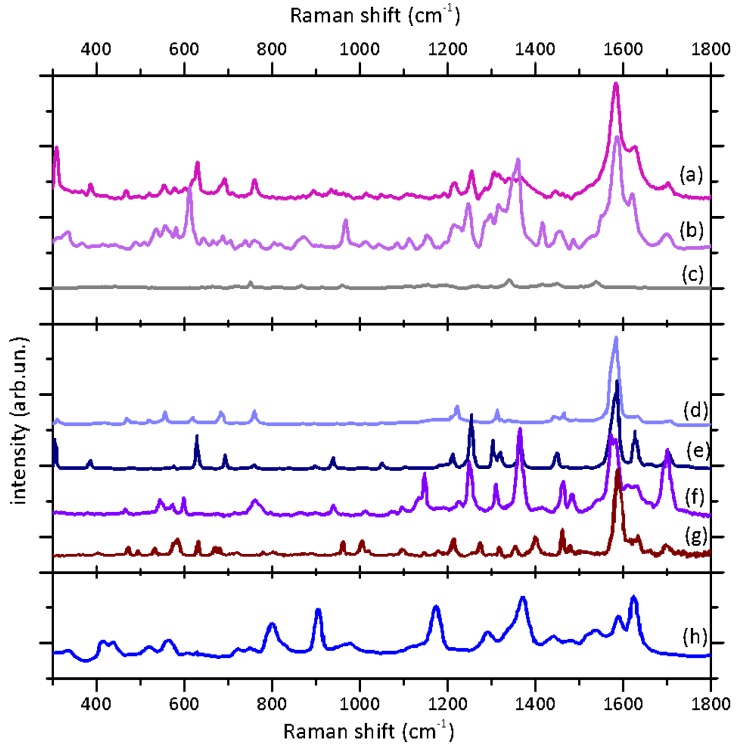
Raman and SERS spectra: results and literature references. Top panel: (**a**) the Raman spectrum acquired on the colored area of the archeological sample and (**b**) the SERS spectrum of the DMF extract from the sample; (**c**) the SERS spectrum of DMF is displayed for reference. Central panel: (**d**) Raman reference spectra of 6-monobromoindigotin (ref. [36], excitation laser line: 785 nm); (**e**) 6,6’-dibromoindigotin (ref. [35], excitation laser line: 633 nm); (**f**) indigotin (ref. [35], excitation laser line: 633 nm); (**g**) indirubin (ref. [40], excitation laser line: 532 nm). Bottom panel: (**h**) the SERS reference spectrum of the DMF extract of indigo acquired using the Lee–Meisel Ag colloid as a SERS substrate [39].

**Figure 4 molecules-25-01417-f004:**
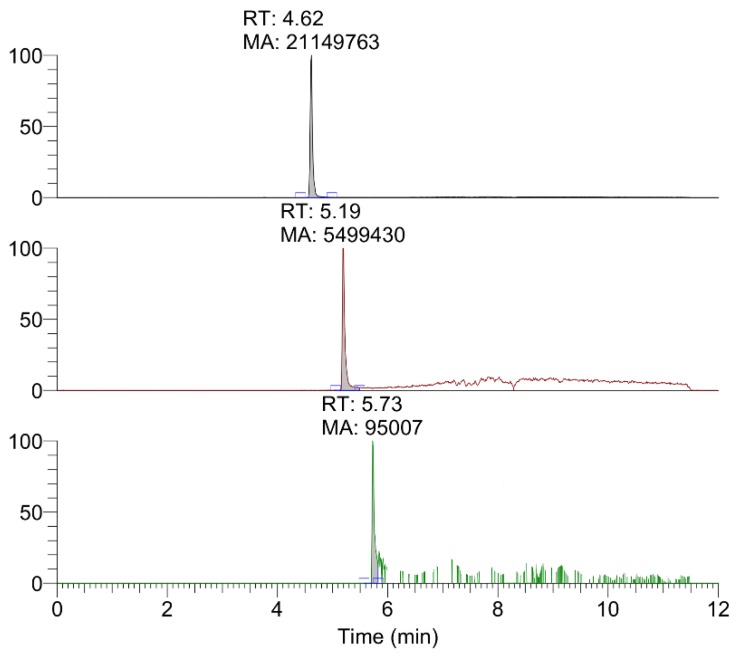
HPLC-HRMS chromatograms relative to non-brominated (top), monobrominated (middle), and dibrominated (bottom) indigoids in the sample DMSO extract.

**Table 1 molecules-25-01417-t001:** Comparison of the main peaks in the Raman spectrum of the sample with the literature ones for indigotin (IND, [35]), indirubin (INR, [40]), 6-bromoindigotin (MBI, [36]) and 6,6’-dibromoindigotin (DBI, [35]) together with their relative tentative assignations based on the literature.

Assignation	Sample	IND (633 nm)	INR (532 nm)	MBI (785 nm)	DBI (633 nm)
δ(C=C), δ(C=O)			210		
ρ(CH), γ(C=C)_2_	251	254			
γ(C-C=C)	274	266		267	
γ(CC), ν(C=O), ν(CBr)	309			304	306
γ(CC), ν(CBr)				364	
γ(CC)	387				386
	468	466	471	466	
δ(C=C-CO-C)			496		
ρ(CH), ρ(NH)	519			516	
δ(CC), δ(CH), δ(NH)			532		
ρ(CH), ρ(CO)	553	545		554	
δ(C=O), δ(CNHC)	578		584		574
δ(C=O), ν(CC-C-C-CN)	602	599			
ν(C-C-C), ν(CC-CO-CC)				619	
γ(NH)	630				629
δ(CC), δ(CH), δ(NH)			631 669		
δ(CC), ρ(CH), ρ(CO		674	680	682	
δ(CH), δ(NCC)	691				692
ν(HN-CC-CO)	712			716	
δ(CH), δ(NCC)	760	762		750	757
γ(CH), δ(CC) ring, ν(CC), ν(CN)				868	
γ(CH), ν(CC), ν(CN)	893			894	896
δ(CH), ρ(CH)	935	940		940	937
δ(CC), δ(CH)			962		
ν(CC), δ(CC)	1015	1013	1006	1011	
ν(CC), γ(CH)				1046	1048
δ(CH)		1096	1097	1125	
δ(CC), δ(CH)		1135			
		1148			1161
	1217	1224	1214	1219	1211
ν(CC), δ(CH), δ(C=O)	1255	1250		1244	1252
δ(CH), δ(NH), ν(CC)			1273		
ν(CC), ρ(NH-C=C-NH)	1306	1310		1308	1301
	1316		1317		1316
ρ(NH-C=C-NH)	1340			1337	
δ(NH), δ(CH), ρ(CH)	1350		1354		
	1368	1364		1362	1364
	1446		1460	1441	1446
	1462	1462	1479	1461	
ν(C=C)		1573			
	1583	1581	1587	1580	1582
		1613	1634		
ν(C=O), ν(C=C)	1628	1628		1625	1625
ν(C=C)		1659			1657
ν(C=O), ν(C=C)	1702	1701		1700	1700

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
