# Peer review of "Dyes from the Ashes: Discovering and Characterizing Natural Dyes from Mineralized Textiles"

_molecules, 2020, doi:10.3390/molecules25061417_

Round 1

Reviewer 1 Report

MOLECULES-682536

"Dyes from the ashes: discovering and characterizing natural dyes from mineralized textiles"

This is a very interesting project, in which authors aim to characterize the purple color found in a "mineralized textile fragment" in the Archeological Museum of Naples (MANN). The authors aim also at identifying the sea snail species (mollusks species) used to dye this fragment.

However, there are several fundamental issues that must be carefully addressed for this paper to be accepted in Molecules.

i) In the introduction, authors must describe how it is possible to identify the species used to dye a textile as well as the limitations of this approach (consider that you are extracting the dye from the textile, and not from the murex). In its present form, it is only vaguely stated that is was done in literature. Please explain precisely the data necessary to assess the species used to dye an historical textile (analyzing extracts from the dyed textile). Why is this important? Because, it will be very difficult to ascertain accurately the species used, unless you have data on the original dyeing methods. Adding to this, it would be necessary to know or to predict the ageing of the textile: this issue is never addressed in this manuscript.

Authors should be aware that the bromine bond may be cleaved in certain conditions, such as exposure to sun, and certainly to high temperatures. Scission of the bromine bond will lead to mono-bromo derivatives and indigo.

ii) Raman spectra: authors must add to their Figure 3, the SERS spectra of indigo, indirubin, 6,6'-dibromoindigo and a monobromo derivative. The main peaks attributed to CBr, NH, CNC, CNO for the various compounds should be added to the spectra.

A table with the more relevant bands should be prepared for each of these compounds. This table will allow the reader to compare data published in the literature with the data obtained in this work.

iii) If authors could display a DAD chromatogram it would be important, as quantification of teh chromophores is more easy to perform.

iv) It is essential to separate indigo (that the authors refer as indigotin) from indirubin. If this is not carried out, the paper cannot be accepted. This is very straightforward using HPLC-DAD.

v) How was quantification performed by HRMS? By comparison with calibration curves using standards? Please add this information into the experimental sections.

The information on how samples were extracted for HPLC-HRMS analysis must be completed.

2 mg of sample were dissolved in xxx ml of DMSO and heated at 70 ◦ C for 5 min ? How was the extract filtered, prior to be suspended in methanol?

IN SUMMARY. The paper, including the abstract, should be rewritten taking into consideration the points raised in i)-v). In its present form the abstract is vague and does not allow the reader to assess the novelty of the work and to know what were the main results obtained and the advances in science.

Other minor points that should be clarified and corrected are described below.

1) Please, use purple for naming the color throughout your manuscript (and not violet)

2) line 46: first time MAAN is used, please give its full name; explain how it is know that it is a "luxury fabric"?

3) line 50: please, add reference to "taking into account of the current scientific discussion"

4) line 54: "identification of the purple dyed sources"

5) line 59: please, add reference

6) line 65: please, add reference; consider, for example, "La porpora. Realtá e immaginario di un colore simbolico". Or other that you consider more suitable.

7) lines 70-75: please, explain precisely in which conditions and how it is possible to make "the identification of shellfish purple "

8) line 81: please summarize the information on references 24 and 25 that allow to conclude that "the archaeological evidence suggests that this fabric may be regarded as an example of purple and gold tapestry "

9) In Figure 2 can you please help the reader to see "z-twisted in an angle of 35-50°", by drawing in a part of the SEM image, for example. It is based on this SEM image that it is concluded that "the mineralized textiles found in Pompeii has been made of wool fibers"?

This is stated in line 104, in which it is also concluded that the textile "dyed in shellfish purple". This conclusion is not possible to be reached based on SEM analysis: please correct

10) Figure 4 should be prepared to be published in a scientific journal; as it is presented now, it would be acceptable for a report

Author Response

Please see the attachment (see reviewer 1).

Reviewer 2 Report

Review of manuscript "Dyes from the ashes: discovering and characterizing  natural dyes from mineralized textiles" by authors Alessandro Ciccola, Ilaria Serafini, Francesca Ripanti, Flaminia Vincenti, Francesca Coletti, Armandodoriano Bianco, Claudia Fasolato, Camilla Montesano, Marco Galli, Roberta Curini and Paolo Postorino

The article does not present any major innovative aspect from the scientific point of view. Nevertheless, it represents a significant case study mostly because of the relevance of the archaeological site and what the findings represent from a cultural point of view.

Moreover, some major differences between the aims presented in the introduction/abstract and the results/findings highlighted in the the discussion/conclusion make the overall aim of the manuscript unclear. This aspect should be revised to make the article acceptable for publication. Other modifications (sentences rephrases, clarification of techniques and findings, ...) would also be required before considering publication of the manuscript. Please, refer to the annotated PDF for more detailed remarks.

Taking into consideration the aforementioned remarks, I would recommend submitted to a more archaeologically-focused journal such as Archaeometry, Journal of archaeological science reports or Archaeological and Anthropological Sciences rather than Molecules. 

Author Response

Please see the attachment (see reviewer 2).

Reviewer 3 Report

This work presents the study of a unique purple mineralized textile found in Pompeii. Raman results showed the presence of Tyrian purple as the main colorant resposible to give its characteristic color and thaks to chromatographic results it was determined the relative contents of different mono-, di- or non brominated compounds. From my point of view this last approach is the most relevant part of the work, showing that the used dye is related mainly to non brominated compounds. However, some minor revisions should be perfomed before the publication of this paper.

First, I want to remark the good interpretation of the Raman and SERS results that led to discussion of the presence of the different brominated compounds and the subsequent chromatographic analysis,.

In my opinion, the most relevant and novel part of the work is the chromatographic result and its discussion. However, the 3 chromatograms presented in this paper should be eddited in a better way, such as:

Add the x axis scale (retention time). There is lot of information useless, mainly in the right part (probably because you just copied and pasted the chromatograms from the software). The peaks cannot be seen easyly, only the main peak. I encourage to perform a good quality chromatograms to present in the paper, such as the one attached, with the retention times over each peak, and starting from minute 0, without empty spaces that only difficult the observation of peaks.

Moreover, you do not explain the procedure that gives to the quantitative results of 80% indigotin/indirubin, 19% monobromo and 1% dibromo compounds. How did you do it? An explanation of how you get these contents is crucial since these results are the most novel issue of the work. Please add it.

Round 2

Reviewer 2 Report

Review of reviewed manuscript “Dyes from the ashes: discovering and characterizing 3 natural dyes from mineralized textiles” by Alessandro Ciccola et al.

The authors addressed all of the remarks from the reviewers. All of the authors edits made the manuscript much easier to read and to follow and highlighted better the aims and findings of the study.

 Some minor comments can be found below:

Abstract

  • Rephrase: “resulted fundamental to progressively increase the analytical information”

Introduction

  • Line 49: did the authors mean “value and the price”?
  • Line 53: “the different color shades range”
  • Line 55: “different purple hues”
  • Line 58: “Currently, researches”
  • Lines 60: several in-depth researches” in order to be in line with the “Currently, researches” form line 58
  • Line 63: “final shade colors”
  • Line 73: according to the reported
  • Line 79-80: rephrase “it has to be taken into account of dyeing procedure” in something like “the dyeing procedure has to be taken into consideration”
  • Line 93: is to discuss

Results

  • Lines 112-113: rephrase: “it has been fundamental the prior evaluation of the material preservation”. The sense of this sentence is very unclear
  • Line 118: “some considerations”

Discussion

  • Line 223 “has occurred”
  • Line 234: “To prove”
  • Line 237: “no affection” should be replaced. This is not right in this context.
  • Line 242: “appropriate to consider” rather than “appropriate to considerate”?
  • Line 250 and 251: “relative quantification was performed by comparing the area of the peaks of interest to the sum of the area of all the peaks” was stated previously. This should be removed or rephrased differently to avoid repetition
  • Line 274: “this mollusc is naturally present as both as indigo-poor and as indigo-rich exemplars, and the latter”. Please rephrase.

Author Response

Dear Editor,

We would thank you and the referee for the revision and for the appreciation about the improvements and the corrections we applied to the paper in the previous revision.

We have revised the manuscript taking into account of the final suggestions provided by the reviewer. In the text of the revised paper (please see the attachment), the major changes, made on the base of the reviewers’ comments, are highlighted in green.

In the following lines, the reviewers’ original comments have been reported in italic; we followed the reviewer suggestions and the changes are provided in the answers (in red). In some cases, some reviewer comments are grouped, because they are only ortographic or typographical mistakes.

Thank you for your consideration.

Yours sincerely,

Paolo Postorino

Point 1:

Rephrase: “resulted fundamental to progressively increase the analytical information”

Response 1:

We rephrased the sentence in “resulted fundamental to maximize the information content” (Line 25).

Point 2:

Line 49: did the authors mean “value and the price”?

Response 2:

“Vale” was modified in “value” (Line 49), as we meant.

Point 3:

Line 53: “the different color shades range”

Line 55: “different purple hues”

Line 58: “Currently, researches”

Lines 60: several in-depth researches” in order to be in line with the “Currently, researches” form line 58

Line 63: “final shade colors”

Line 73: according to the reported

Response 3:

All the suggestion, related to orthographical errors, were accepted (the Lines correspond to those mentioned by the Reviewer).

Point 4:

Line 79-80: rephrase “it has to be taken into account of dyeing procedure” in something like “the dyeing procedure has to be taken into consideration”

Response 4:

We modified the sentence in: “the applied dyeing procedure has to be taken into consideration” (Lines 79-80).

Point 5:

Line 93: is to discuss

Response 5:

We corrected the expression (Line 93).

Point 6:

Lines 112-113: rephrase: “it has been fundamental the prior evaluation of the material preservation”. The sense of this sentence is very unclear

Response 6: 

We rephrased the sentence in: “it has been fundamental to evaluate the preservation state of the fibers, in order to obtain preliminary information from its optical appearance” (Lines 112-114).

Point 7:

Line 118: “some considerations”

Line 223 “has occurred”

Line 234: “To prove”

Response 7:

We accepted all the corrections (Line 119, 224, 234).

Point 8:

Line 237: “no affection” should be replaced. This is not right in this context.

Response 8:

We modified the sentence in: “no interference from colloid signals” (Line 238).

Point 9:

Line 242: “appropriate to consider” rather than “appropriate to considerate”?

Response 9:

We modified the expression according the Reviewer suggestion.

Point 10:

Line 250 and 251: “relative quantification was performed by comparing the area of the peaks of interest to the sum of the area of all the peaks” was stated previously. This should be removed or rephrased differently to avoid repetition

Response 10:

We modified the sentence in “and semi-quantitative analysis was performed according the method discussed in Section 2.3 (Line 251).

Point 11:

Line 274: “this mollusc is naturally present as both as indigo-poor and as indigo-rich exemplars, and the latter”. Please rephrase.

Response 11:

We changed the sentence in: “in nature there are indigo-poor and indigo-rich exemplars of this mollusc species, and the latter ones are likely the original matrix” (Lines 274-275).